# Association Between Income and Well-Being Among Working Women in Japan

**DOI:** 10.3390/healthcare13030240

**Published:** 2025-01-24

**Authors:** Takao Suzuki, Kiriko Sasayama, Etsuko Nishimura, Noyuri Yamaji, Erika Ota, Eiko Saito, Daisuke Yoneoka

**Affiliations:** 1Graduate School of Public Health, St. Luke’s International University, Tokyo 104-0045, Japan; 24mp217@slcn.ac.jp; 2Sustainable Society Design Center, Graduate School of Frontier Sciences, The University of Tokyo, Chiba 277-0882, Japan; 3Faculty of Nursing, Komazawa Women’s University, Tokyo 206-8511, Japan; 4Institute of Clinical Epidemiology, Showa University, Tokyo 142-8555, Japan; 5Global Health Nursing, Graduate School of Nursing Science, St. Luke’s International University, Tokyo 104-0044, Japan; 6Center for Surveillance, Immunization, and Epidemiologic Research, National Institute of Infectious Diseases, Tokyo 162-0052, Japan

**Keywords:** working women’s health, well-being, income, Japan

## Abstract

**Background**: Income is a key determinant of well-being; however, its effects are often nonlinear. In Japan, working women face unique limitations to their well-being, including substantial gender wage gaps, caregiving responsibilities, and female-specific health conditions. This study aimed to investigate the association between income and well-being, with a focus on potential nonlinear patterns and effect modification with various factors. **Methods**: A nationwide survey of 10,000 working women aged 20–64 years was conducted in Japan in 2023. Well-being was assessed using four items from the Office for National Statistics-4 questionnaire, each rated on a 0–10 Likert scale. Tobit regression models were used to assess the association between household income and well-being after adjusting for demographic, socioeconomic, and health-related factors. **Results**: Annual household income was positively associated with well-being in women earning up to 8–10 million JPY annually, beyond which the effect was attenuated. Women with mental health issues or insomnia reported significantly lower well-being scores regardless of their income level (*p* < 0.05). Marital status and caregiving responsibilities had moderate effects, whereas having more children diminished the positive effect of income among higher-income households earning over 8 million JPY annually. **Conclusions**: This study highlights the need for integrated policies that address both economic disparities and health-related challenges to improve the well-being of working women in Japan. Targeted interventions focusing on female-specific health conditions are particularly important.

## 1. Introduction

Income has long been recognized as a primary factor influencing well-being [1,2]. However, traditional policy frameworks have predominantly focused on economic indicators such as Gross Domestic Product, often neglecting the complex, multidimensional aspects of individual quality of life and well-being [3,4]. Recently, research and policy perspectives have shifted to emphasize subjective well-being, a holistic measure encompassing psychological health, social relationships, and overall life satisfaction, as essential for understanding true societal progress [5]. Although income contributes markedly to well-being, its impact is nonlinear. The “Easterlin Paradox” demonstrates that while income initially increases happiness, additional financial gains yield diminishing returns [6]. Similarly, emotional well-being plateaus when income reaches a level sufficient to cover basic needs and personal aspirations [7]. These findings highlight the intricate role of income in well-being and emphasize the need for policies that consider a broader set of well-being determinants beyond financial metrics.

Enhancing well-being benefits not only individuals, but also society at large. Higher well-being is associated with improved physical health, enhanced cognitive functioning, and greater resilience to stress. Individuals with high life satisfaction live longer and have a lower risk of cardiovascular diseases than those with low life satisfaction [8]. Happiness is also positively associated with workplace productivity and professional growth; happier individuals tend to be more motivated and collaborative, fostering economic independence and higher income potential [9]. On a societal scale, higher well-being promotes stronger community bonds, reduces healthcare costs, and improves labor productivity [10]. For example, the 2021 World Happiness Report highlighted that countries with elevated happiness levels demonstrated greater resilience and compliance during the COVID-19 pandemic, enabling quicker economic recovery and more effective public health responses [11]. However, despite being a highly developed nation, Japan ranks 51st out of 143 countries in the 2024 World Happiness Report, the lowest among the G7 nations [12]. Particularly concerning is Japan’s ranking of 110th in the Dystopia Happiness Score, which evaluates fundamental happiness metrics.

Japan faces unique challenges in translating economic gains into well-being, particularly for working women. Prolonged deflation and stagnant economic growth have resulted in slower income increases than in other developed nations. Although women’s labor force participation rate has increased, a persistent gender-related income gap still exists. A 2022 survey by the Organisation for Economic Co-operation and Development (OECD) showed that Japanese women’s average income was 78.7% of men’s, significantly below the OECD average of 87.9% across its 38 members [13]. This disparity impedes women’s financial independence and perpetuates social inequality. Gender-related income inequality also affects well-being. For example, perceptions of unfairness negatively affect individuals’ overall happiness in the United States [14]. Moreover, working women in Japan face unique health challenges, including menstrual disorders, menopause, endometriosis, and polycystic ovary syndrome. Endometriosis can reduce labor productivity and learning efficiency in women, contributing to a lower quality of life [15]. Menopause and reproductive health disorders also play a crucial role, and these health issues exacerbate stress and decrease well-being [16].

The situation of women in Japan is deeply rooted in long-standing historical and cultural factors. Traditionally, Japan has adhered to a gendered division of labor, where men work outside the home and women take care of household duties [17]. This division of labor can be attributed to historical factors, as well as deliberate government policies during the formation of the Japanese-style welfare state in the postwar period. The Japanese-style welfare state was characterized by a system where corporations and families played central roles in providing welfare services, while the government’s role was limited to providing minimum living standards [18]. This system was supported by a labor system based on seniority and lifetime employment during the period of high economic growth, with corporations offering comprehensive welfare benefits to their employees in return. Moreover, given Japan’s traditional emphasis on the roles of family and community, mutual aid within families and among community members played a significant role in welfare. In other words, Japan’s postwar social security system was built on the premise of a gendered division of labor, where husbands worked for corporations and wives took care of household chores and child-rearing. This approach has been criticized as neglecting welfare, and past Japanese policies must be strongly condemned [18]. The Japanese welfare system, constructed on the assumption that women would be responsible for childcare and caregiving, was implemented by the government. This system is now dysfunctional due to changes in women’s social participation, family structures, and community functions. Despite the need for improvement, the concept of the gendered division of labor persists. A 2022 Cabinet Office survey on gender equality revealed that a significant majority, 78.8%, of respondents believed that men are still favored in society, compared to only 14.7% who believed that the status of men and women has become equal. Moreover, 33.5% of respondents agreed with the notion that “husbands should work outside the home and wives should stay at home,” indicating that traditional gender roles remain prevalent despite increasing female labor force participation [19]. When respondents were asked if they agree that women’s heavy involvement in childcare, caregiving, and housework hinders their career advancement, a striking 84.0% of women affirmed this view [19]. The situation of women in Japan is a complex interplay of historical factors and government policies. The traditional gender-based division of labor, where men are expected to work outside the home and women to take care of domestic responsibilities, is now dysfunctional and a major hindrance to women’s advancement in society. To address this issue, the government must implement more proactive policies to achieve gender equality.

To address these interconnected dynamics, this study examined the association between household income and well-being among working women in Japan, focusing on female-specific health conditions as potential effect-moderating factors. By exploring these factors, this study aims to provide actionable insights for policies to support women’s well-being through both economic and health-focused interventions.

## 2. Materials and Methods

### 2.1. Study Design and Questionnaire

Participants were recruited from a nationwide panel of Japanese women residing across all 47 prefectures between 28 February and 7 March 2023, aged 20–64 and sufficiently fluent in Japanese to respond to the survey. ”Working women” were defined as those employed under a contract and receiving pay at the time of the survey, regardless of the job type or working hours. As of June 2022, the panel provider has maintained an active pool of 5.24 million individuals, including men and women, in Japan. A target sample of approximately 10,000 women was set to enable a detailed analysis according to prefecture. The sample size distribution across age groups (20s, 30s, 40s, 50s, and 60–64 years) was calculated using a baseline well-being score from the 2021 World Value Survey [11]. A quota-based random sampling approach was applied to ensure a nationally representative sample, by aligning age and regional proportions with data from the 2015 National Census. Participation was on a first-come, first-serve basis, closing once the targeted quotas for each age group and prefecture had been satisfied. Respondents were required to complete all survey items to prevent missing data.

The questionnaire was created after an extensive review of the existing literature on well-being and epidemiological studies [20]. The survey items were organized into three primary sections—health, sociodemographic characteristics, and psychological factors—with a particular focus on well-being. Well-being questions were adapted from the UK Office of National Statistics (ONS) [21]. Well-being was assessed using four Likert scale questions. Participants rated their well-being through four items, ONS1: life satisfaction, ONS2: feeling that things carried out in life are worthwhile (i.e., worthwhile), ONS3: happiness, and ONS4: feeling anxiety (i.e., anxiety) on a scale from 0 to 10, where 0 represented “not at all” and 10 represented “completely.” The detailed sampling scheme and questionnaire are available and were previously published [20]. Ethical approval was granted by the Ethics Committee of St. Luke’s International University (approval number: 22-A089) on 21 December 2022. Online informed consent was obtained from all participants following a full explanation of the study purpose, methods, and secondary use of the data. 

### 2.2. Statistical Analysis

Baseline data were reported as the mean (standard deviation [SD]) or proportion (%). Analysis of variance (ANOVA) was used for the continuous variables, and a Chi-squared test was used for categorical variables to determine the difference in baseline characteristics between income categories (<2, 2–4, 4–6, 6–8, 8–10, 10–12, 12–14, and ≥14 M JPY). To visualize the relationship between household income and well-being, the average well-being scores according to income categories and 95% confidence intervals (CIs) were plotted and stratified by basic characteristics and six health conditions, including problems related to menstruation (menstrual problems), mental health problems/insomnia, anemia, headache, gastrointestinal problems, and eating/feeding problems. These six health conditions were selected because they are among the top six female-specific health conditions faced by working women in Japan [20]. The differences in each category were statistically tested, using the *t*-test for two categories or ANOVA for three categories. Finally, the following (Type 1) Tobit regression models censored at 0 and 10 were applied to estimate the association between income and well-being after adjusting for basic characteristics, health conditions, and their interactions.$$y_{i}^{*}=\alpha_{0}+\alpha_{1}Income_{i}+\sum_{j=1}^{9}\beta_{j}x_{ij}+\sum_{j=1}^{9}\gamma_{j}{Income_{i}\times x}_{ij}+\sum_{j=1}^{4}\delta_{j}z_{ji}+\varepsilon_{i},\;\varepsilon_{i}∼N\left(0,\;\sigma^{2}\right),\;y_{i}=\left\{\begin{array}{c} 0\;if\;y_{i}^{*}<0\\ y_{i}^{*}\;if\;{0\leq y}_{i}^{*}\leq 10\\ 10\;if\;y_{i}^{*}>10, \end{array}\right.$$
where $y_{i}$ is the well-being score of *i*th individual, $y_{i}^{*}$ is an unobserved (latent) variable, $x_{ij}$ is the covariate including age groups, dummy variables of marital status, providing in-home care, number of children, and the above six health conditions of *i*th individual, $z_{ji}$ is the confounders, including occupation, residential region, final academic background, and working shift style. α, β, γ, δ, and σ are regression parameters. $\sum_{j=1}^{9}\beta_{j}x_{ij}$ indicates the shift in the curve between income and well-being and $\sum_{j=1}^{9}\gamma_{j}{Income_{i}\times x}_{ij}$ indicates the change in the slope of the curve. Then, to check the robustness of our results, we have calculated the bounding values for the income coefficients, following the methodologies proposed by Altonji et al. (2005) and Oster (2019) [22,23]. The bounding value is $\beta^{*}=\tilde{\beta }-\frac{\left(\dot{\beta }-\tilde{\beta )}(R_{max}^{2}-\tilde{R^{2}}\right)}{\tilde{R^{2}}-\dot{R^{2}}}$, where $\dot{\beta }$ and $\dot{R^{2}}$ are, respectively, the point estimate and R-squared value for (simplified) linear regression with only income covariates, $\tilde{\beta }$ and $\tilde{R^{2}}$ are analog values for linear regression with all covariates in the above regression model, and $R_{max}^{2}=min(1,\Pi {\tilde{R}}^{2})$ with Π=1.3 or 2.0, as with Dantas (2023) [24]. Statistical analyses were conducted using STATA (Stata Corp, College Station, TX, USA, version 18.0) and R (R Foundation for Statistical Computing, Vienna, Austria, version 4.3.1) software. For two-sided statistical tests, a *p*-value of <0.05 was considered statistically significant.

## 3. Results

Participants’ main characteristics are listed in Table 1 and Appendix A. A total of 10,000 working women were enrolled in the study between 28 February and 7 March 2023. Out of the 10,000 participants, 1103 (11.0%) and 378 (3.8%) working women were categorized into the lowest (<2 M JPY) and highest (≥14 M JPY) categories of household income, respectively. The average age was 44 (SD: 11.6), 43 (SD: 12.0), 43 (SD: 11.4), 43 (SD: 11.1), 43 (SD: 10.9), 43 (SD: 10.9), 44 (SD: 9.9), and 46 (SD: 10.7) years for the household income categories of <2, 2–4, 4–6, 6–8, 8–10, 10–12, 12–14, and ≥14 M JPY, respectively. The average age of overall participants was 43 (SD: 11.4) years old. High school graduation was the most common final academic background (39.1%) among those with the lowest household incomes. In contrast, a university degree was the most common final academic background (56.1%) among the group with the highest household income.

Figure 1 shows the association curves between the average well-being scores and household income groups stratified according to baseline characteristics. Across all age groups, ONS1–3 displayed similar curve shapes up to an income level of 10 M JPY (Figure 1B). Beyond this point, the variance increased among the age groups, indicating that higher income levels affected them differently. Notably, high ONS4 values were consistently observed in younger individuals (aged ≤ 30 years), regardless of income level. A significant shift in the ONS1–3 curve shape was observed regarding marital status (Figure 1C, *p* < 0.05) and provision of in-home care (Figure 1D, *p*-values ranged from 0.150 to <0.001), with differences increasing as income increased. The number of children did not substantially alter the curve shapes for ONS1–3, indicating relative consistency regardless of family size (Figure 1E, *p*-values ranged from 0.896 to <0.001). However, mothers with three or more children exhibited a decrease in the ONS1–3 values in the income group ranging from 8 to 12 M. These tendencies were also observed even when participants were stratified by part- or full-time employment (Figures 3–6). More detailed *p*-values are shown in Appendix A.

Figure 2 shows the association curves between the average well-being score and household income groups stratified according to the six health conditions. Individuals with female-specific health conditions had lower well-being scores in ONS1-3 than those without such health conditions. Notably, the curves (except for anemia) showed a significant parallel shift up to an income level of 10M JPY (average *p*-value was 0.032); however, this significant difference decreased for income levels above 10M JPY (average *p*-value was 0.318). The most substantial curve shifts were observed in individuals with mental health problems or insomnia, with an approximate decrease at one point in the ONS score (Figure 2B, (average *p*-value was 0.005)). This result was confirmed by regression analysis (Table 2). Furthermore, regarding anemia (Figure 2C), lower-income groups (≤6M JPY) experienced a more significant decrease in well-being than higher-income groups, leading to a larger difference between those with and without anemia (average *p*-value was 0.035).

Figure 3, Figure 4, Figure 5 and Figure 6 show the association curves between the average well-being score and household income groups stratified according to baseline characteristics, the six health conditions, and employment status. Here, we observed a similar trend even after stratifying by the employment status:, i.e., across all age groups, ONS1–3 displayed similar curve shapes up to an income level of 10M JPY. Beyond this point, the CIs became large due to the small sample size (Figure 3 and Figure 5). Despite the employment status, individuals with female-specific health conditions had lower well-being scores in ONS1-3 than those without such health conditions (Figure 4 and Figure 6). In addition, Figure 7, Figure 8, Figure 9 and Figure 10 show similarly stratified curves according to spouse status. Again, we observed similar curve shapes across all age groups in ONS1–3, while the shapes were more different by age groups among those without a spouse (Figure 7 and Figure 9). This tendency was also observed when stratified by female-specific health conditions, with more pronounced differences between subgroups for those without a spouse than for those with a spouse (Figure 8 and Figure 10).

Table 2 presents the estimated regression results of the Tobit model analysis. When examining the main effects, an increase in income resulted in an upward shift in the association curves between income and well-being for ONS1–3, indicating higher well-being scores (0.22 [95% CI: 0.13–0.31], *p* < 0.01 for ONS1; 0.23 [95% CI: 0.15–0.32], *p* < 0.01 for ONS2; and 0.27 [95% CI: 0.17–0.37], *p* < 0.01 for ONS3). In contrast, no significant shift in the association curve was observed for ONS4 with increasing income (*p* = 0.686). Having a spouse led to a significant upward shift in the association curves for all ONS1–4 measures (0.48 [95% CI: 0.24–0.72], *p* < 0.01 for ONS1; 0.49 [95% CI: 0.25–0.73], *p* < 0.01 for ONS2; 0.67 [95% CI: 0.40–0.94], *p* < 0.01 for ONS3; and −0.40 [95% CI: −0.71 to −0.08], *p* = 0.013 for ONS4). However, the presence of a spouse did not significantly alter the slopes of the association curves (*p* = 0.266–0.563 for ONS1–4). In addition, an increase in the number of children caused significant upward shifts in the association curve (*p* = 0.001 and 0.038 for ONS1 and 2, respectively) and changes in the slopes of the association curves for ONS2 (*p* = 0.005) and ONS3 (*p* = 0.024). Mental health problems and insomnia notably shifted the association curves, indicating a worsening of well-being (−1.27 [95% CI: −1.54–0.99], *p* < 0.01 for ONS1; −1.29 [95% CI: −1.56 to −1.03], *p* < 0.01 for ONS2; −1.51 [95% CI: −1.81 to −1.21], *p* < 0.01 for ONS3; and 1.90 [95% CI: 1.55–2.26], *p* < 0.01 for ONS4). In addition, the interaction terms for ONS2 and ONS3 were significant, suggesting changes in the slopes of the association curves (*p* = 0.027 and *p* = 0.043 for ONS2 and 3, respectively). In addition, Appendix A includes the results of the robustness check. For ONS1-3, the bounding values for the income showed the same sign, implicating that the estimated results were robust against the selection, or unobservable to some extent. In contrast, for ONS4, the original coefficient estimates were not statistically significant, and thus the bounding values were also not significantly different from the sign switch (a switch from positive to negative values).

## 4. Discussion

This study examined the association between income and well-being among 10,000 working women in Japan and explored the effect modification of basic characteristics and six female-specific health conditions. The findings indicate that well-being, as measured by the ONS1–3 scores, increases linearly with household income up to approximately 8M JPY. Beyond the income threshold, the positive association was attenuated in most cases, and the variation in responses among the subgroups became more dispersed. In addition, mental health issues and insomnia significantly reduce well-being irrespective of income level, highlighting their profound effect. 

Initially, this study examined the individual characteristics of women, modifying the association between income and their well-being. Providing care not only imposes financial burdens, but also results in various impacts such as the time constraints associated with caregiving or mental stress. The interaction term between income and provision of in-home care revealed that as income increases, women’s well-being tends to improve less (i.e., the slope goes toward zero). In households with higher incomes, women may experience higher workloads and longer working hours in their occupations. Additionally, given the long-standing societal perception in Japan that childcare and caregiving are women’s responsibilities, women continue to bear a disproportionate burden of caregiving, even as women’s social participation has increased. The 2022 Comprehensive Survey of Living Conditions found that 74.5% of primary caregivers are women [25]. The majority of cases, 84.3%, do not primarily rely on businesses for caregiving, but women in households with care recipients tend to shoulder a significant portion of the caregiving responsibilities. These findings suggest that the burden of caregiving imposed on women is so substantial that it cannot be fully compensated for by increased income.

The increase in women’s well-being associated with having more children becomes more pronounced as household income increases. However, in households with an income of 8–12 M JPY, mothers with three or more children experienced a reduction in well-being. Interestingly, the interaction term of the number of children in the regression analysis was significantly negative, suggesting that it flattened the slope of the association curve between income and well-being. This finding implies that as the number of children increases, the curve becomes flatter; thus, the effect of increased income on well-being diminishes gradually. This finding aligns with previous research suggesting that higher household income should offset the negative impact of having children [26]. This phenomenon can be partially attributed to traditional Japanese cultural norms, in which women often stay at home to manage housework instead of pursuing employment outside the household. According to a nationwide household survey in 2022 involving wives under 60 years of age, women spent 247 min on housework on weekdays and 276 min on weekends, approximately four times longer than men, who spent 47 min on weekdays and 81 min on weekends, respectively [27]. In addition, women with high household incomes are more likely to be employed full-time, leaving them with little free time and a substantial burden of housework and childcare. These dynamics suggest that the dual burden of work and home responsibilities, coupled with the unequal distributions of domestic roles, negatively impacts the association between income and well-being. Furthermore, high-income households often allocate more resources to extracurricular education for their children [28], which can impose financial strain and reduce well-being. This study suggests that financial and psychological support is needed regardless of high income, especially for households earning approximately 8–12 M JPY.

Female-specific health conditions also significantly modified the association between income and well-being; an overall decline in ONS1–3 was observed among individuals reporting female-specific health conditions. Except for mental health problems and insomnia, the effects of menstrual problems, anemia, headaches, gastrointestinal problems, and eating/feeding problems showed decreasing disparities between those who had and did not have the problem as income increased. This finding suggests that income can mitigate the negative effects of health conditions on well-being. In contrast, regression analysis showed that mental health issues and insomnia significantly reduced ONS1–3 scores and heightened anxiety, with these effects persisting irrespective of income level. Depression and anxiety disorders have a substantial impact on well-being and impose notable societal costs. For instance, in Japan, annual medical expenditures for depression exceeded 3 trillion JPY in 2008, and women were twice as likely to experience depression as men. According to the Ministry of Health, Labor, and Welfare’s Patient Survey, prevalence rates were consistently higher among women than among men across all age groups. In addition, the negative spillover effects between work and home are significantly associated with poor mental health among working mothers, along with challenges in maintaining a work–family balance [29]. Additionally, women face unique health risks such as premenstrual dysphoric disorder, maternity blues, postpartum depression, and menopausal disorders [30]. To address these issues, workplace support for working mothers is essential, as the psychological and physical burdens of housework, childcare, and employment are health risks unique to women. It is essential to develop and publicize social resources accessible to women. As a case in point, Article 68 of the Labor Standards Act states that “If a woman who finds it to be extremely difficult to work on a day of her menstrual period requests leave, the employer must not make her work on a day of her menstrual period.” However, despite this legal guarantee, the actual utilization rate of menstrual leave is less than 1% [31]. Through laws including the Labor Standards Act, the Act on Equal Opportunity and Treatment between Men and Women in Employment, and The Maternal and Child Health Act, Japan is implementing numerous policies to support women’s advancement in society while addressing the unique health and social challenges faced by women. Especially to promote work–life balance among working women, the Bureau encourages companies to create action plans and set targets based on the Act on Advancement of Measures to Support Raising Next-Generation Children. The companies that successfully achieve their targets are eligible for certification and can display the certification logo on their products. While it is crucial to continue developing new policies, improving the awareness and accessibility of existing government programs is equally important.

Women are inherently more vulnerable to anemia because of regular menstrual bleeding, which necessitates a consistent intake of nutrients [32]. Individuals in low-income groups often have unbalanced diets, leading to a higher prevalence of anemia among these women [33]. In addition to mental health challenges, this study identified significantly lower levels of well-being among low-income groups with anemia. These findings suggest that both low income and inadequate daily nutrient intake are associated with reduced well-being. Dietary habits, a modifiable lifestyle factor, markedly influence mental health and well-being [34]. Therefore, nutritional support targeting low-income groups is necessary to improve the well-being of working women with anemia, along with targeted policies to address poverty, such as increasing women’s wages and expanding child allowance programs. 

This study had several limitations. First, there are concerns about sample size and the sampling method. The sample size, though relatively large, was not equally distributed across all age groups and income brackets. Consequently, the sample sizes for older age groups and higher income brackets were smaller, leading to wider confidence intervals and lower precision. In addition, although quota-based random sampling was employed to ensure representativeness across age groups and regions, the income distribution was not adjusted in the sampling procedure, due to the absence of this information in the sampling panel. However, a comparison of the income distribution of our data with that reported in the 2023 Comprehensive Survey of Living Conditions revealed a high degree of similarity. The national survey indicated an average income of 5,242,000 yen, a median income of 4,050,000 yen, and that 11.6% of households had an annual income exceeding 10 million yen [35]. In our data, the mean and median income fell within the 4–6-million-yen range, while 12.11% of households reported an annual income exceeding 10 million yen. A more detailed comparison of the income class distribution from the 2023 Comprehensive Survey of Living Conditions reveals the following: less than 2 M (21.5%), 2–4 M (27.4%), 4–6 M (19.2%), 6–8 M (12.2%), 8–10 M (8.3%), 10–12 M (4.9%), 12–14 M (2.85%), and 14 M or more (3.9%). This similarity suggests that our data provide a reasonably representative reflection of the population’s income distribution, (partially) ensuring the generalizability to the Japanese population [32]. Second, the survey was conducted online and was self-reported, introducing the risk of potential bias. Selection bias could also have occurred due to online surveys. Furthermore, responses from individuals with a relatively low socioeconomic status, such as older people, those with lower levels of education, those with lower incomes, and those who have limited access to information, are underrepresented. Additionally, while e-commerce site points were offered as an incentive for participation, this might be more effective for lower-income individuals, suggesting that the characteristics of respondents may not fully represent the population. However, despite concerns that the sample might overrepresent individuals with lower household incomes, the examination of the income distribution within the sample revealed that it is, in fact, generally consistent with the population-level distribution in Japan, as discussed above. Therefore, the potential impact of this overrepresentation on the findings is minimal. Self-report surveys are also prone to response bias. Although the ONS4 is a widely standardized measure, the use of a 10-point Likert scale allowed respondents to select neutral responses. Additionally, the presence or absence of symptoms was based on self-reported data, which may not accurately reflect medical diagnosis. Given that respondent characteristics such as education and income can affect the accuracy of responses, these limitations affect reliability. Lastly, while the relationship between maternal well-being and number of children is influenced by the children’s specific characteristics, this study was limited by the absence of detailed data on such factors. Although these characteristics impose certain limitations on the interpretation of the survey results, they implicitly suggest the usefulness of future detailed research.

## 5. Conclusions

This study examined the association between income and well-being among 10,000 working women in Japan, focusing on the effect of employment status, marital status, caregiving responsibilities, and female-specific health conditions. Our research highlights the significant impact of economic hardship and gender-specific health issues on women’s health. As women’s workforce participation continues to rise in Japan and globally, improving work–life balance sustainability becomes increasingly critical. A comprehensive approach is needed that tackles issues at both individual and societal levels. In addition, while this study provides valuable insights into the factors impacting women’s well-being, further research is imperative to deepen our understanding. Specifically, more in-depth investigations are needed to identify the specific mechanisms through which these factors interact and to quantify their relative impact on women’s well-being. Moreover, our research is still insufficient to provide a comprehensive understanding of the causality of the complex interplay of income with family-related variables such as family size, family comorbidities, and living style. Future research should explore these relationships in greater depth, considering factors such as social support networks, access to affordable childcare, and cultural norms that shape parental roles and responsibilities. Also, while this research has focused on working Japanese women, future studies should expand the scope of investigation to include a more diverse population, such as housewives and students. Cross-cultural comparisons would also enrich our understanding of the global impact of female-specific health conditions on well-being.

In conclusion, this study highlights the need for a multi-pronged approach to improving women’s well-being. This approach should encompass not only economic policies, but also social and cultural interventions that address the diverse challenges faced by women in today’s society. We suggest that further research is needed to tackle these issues.

## Figures and Tables

**Figure 1 healthcare-13-00240-f001:**
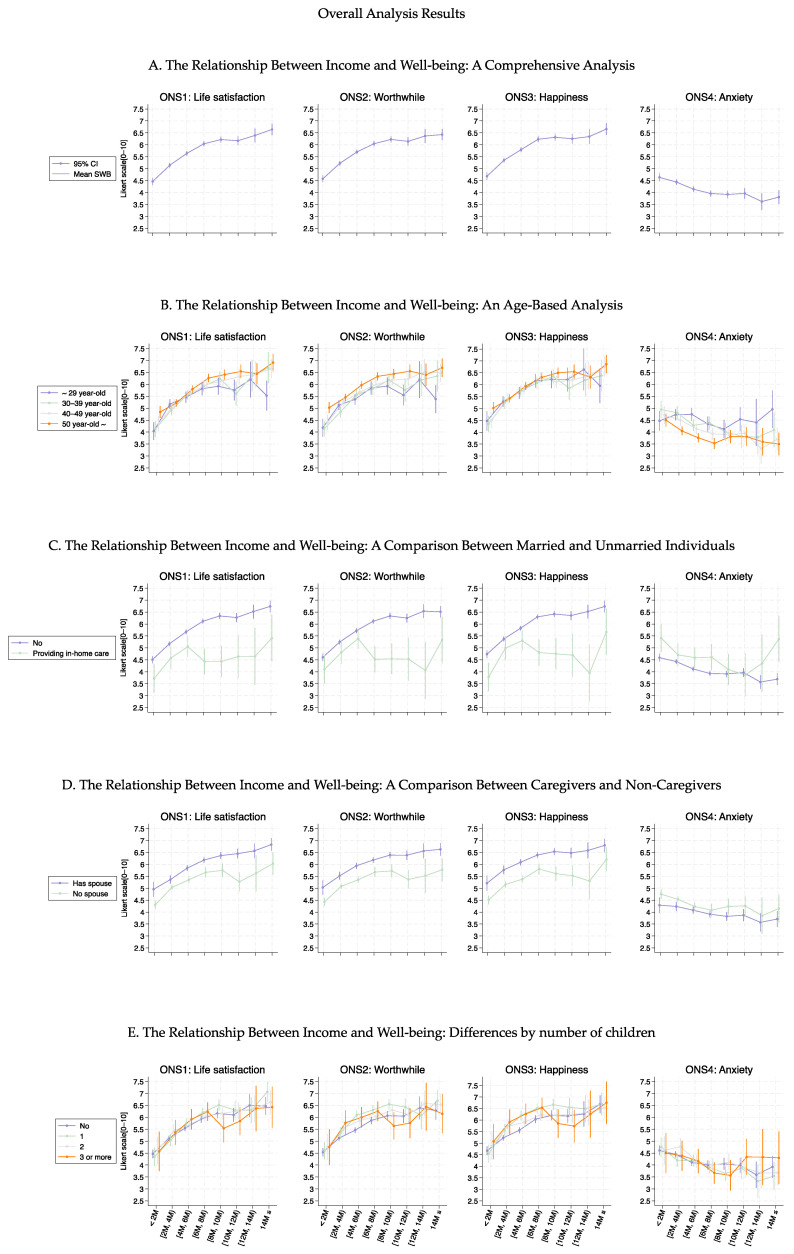
Curves between household income and well-being by basic characteristics.

**Figure 2 healthcare-13-00240-f002:**
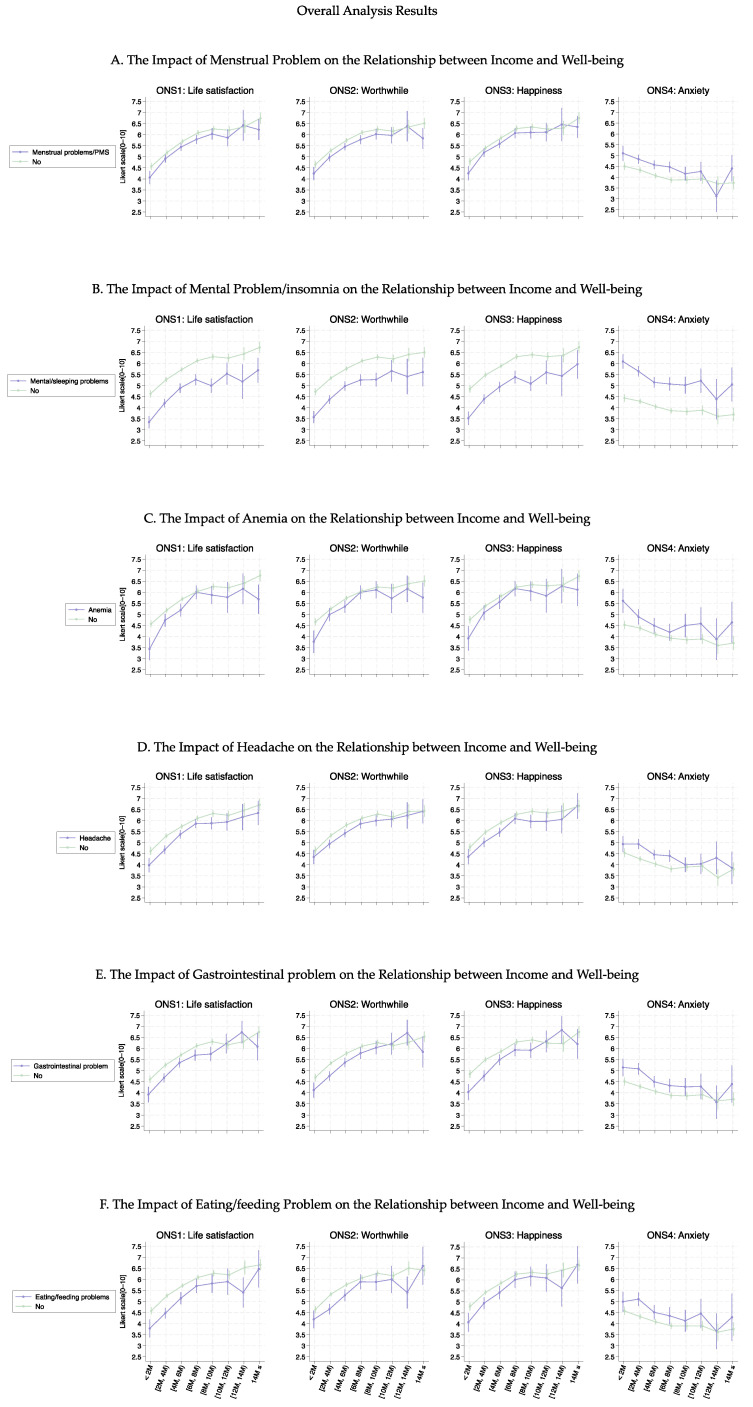
Curves between household income and well-being by female-specific health conditions.

**Figure 3 healthcare-13-00240-f003:**
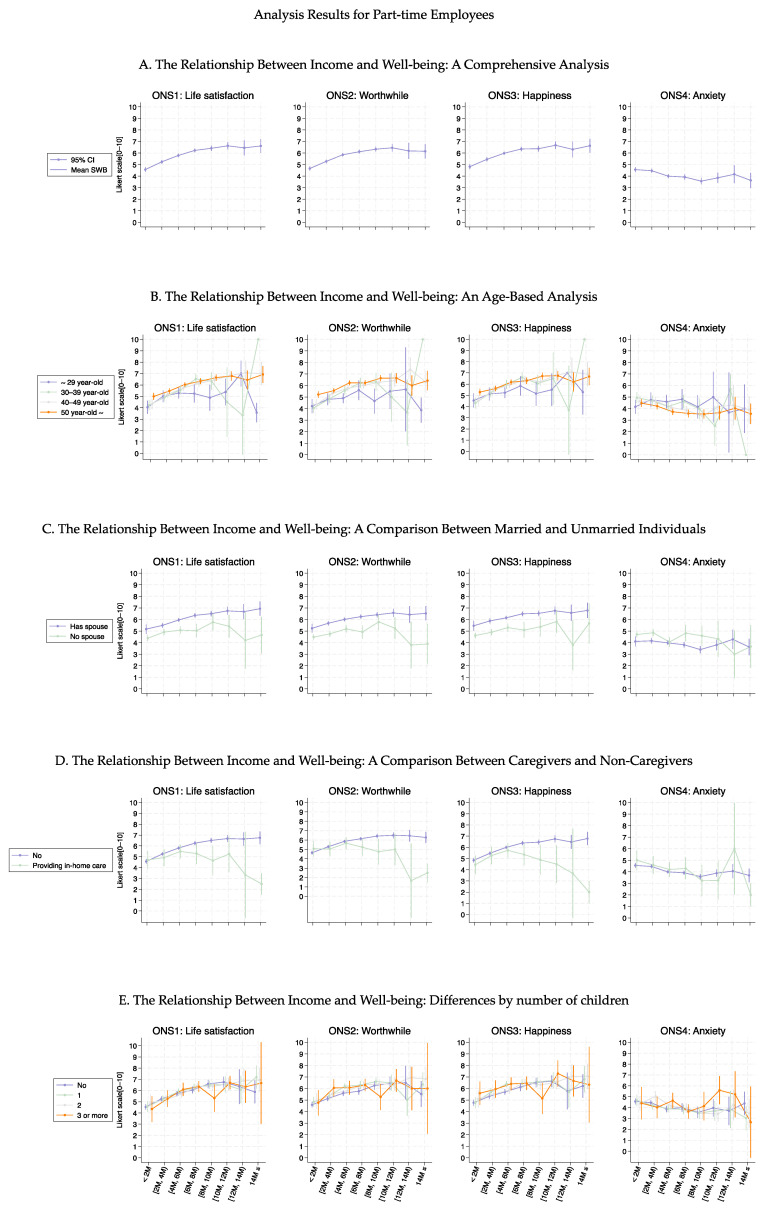
Curves between household income and well-being by basic characteristics among part-time employees.

**Figure 4 healthcare-13-00240-f004:**
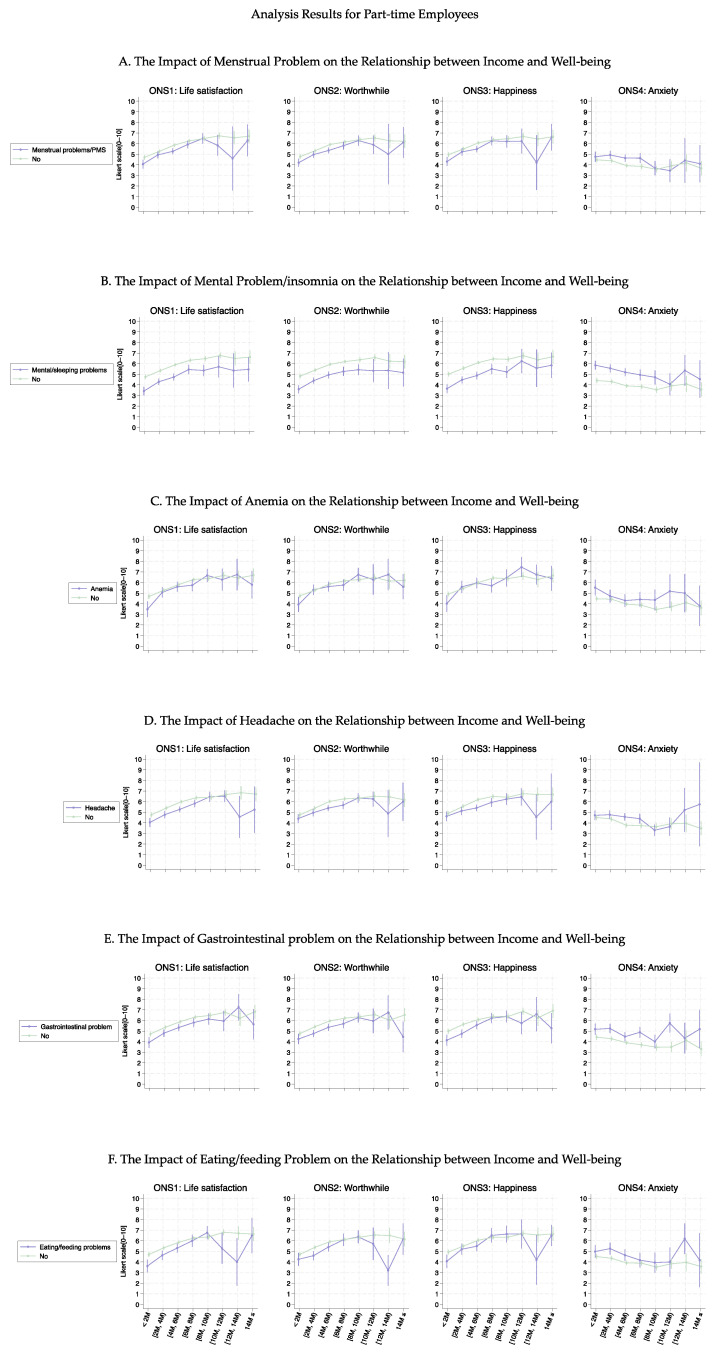
Curves between household income and well-being by female-specific health conditions among part-time employees.

**Figure 5 healthcare-13-00240-f005:**
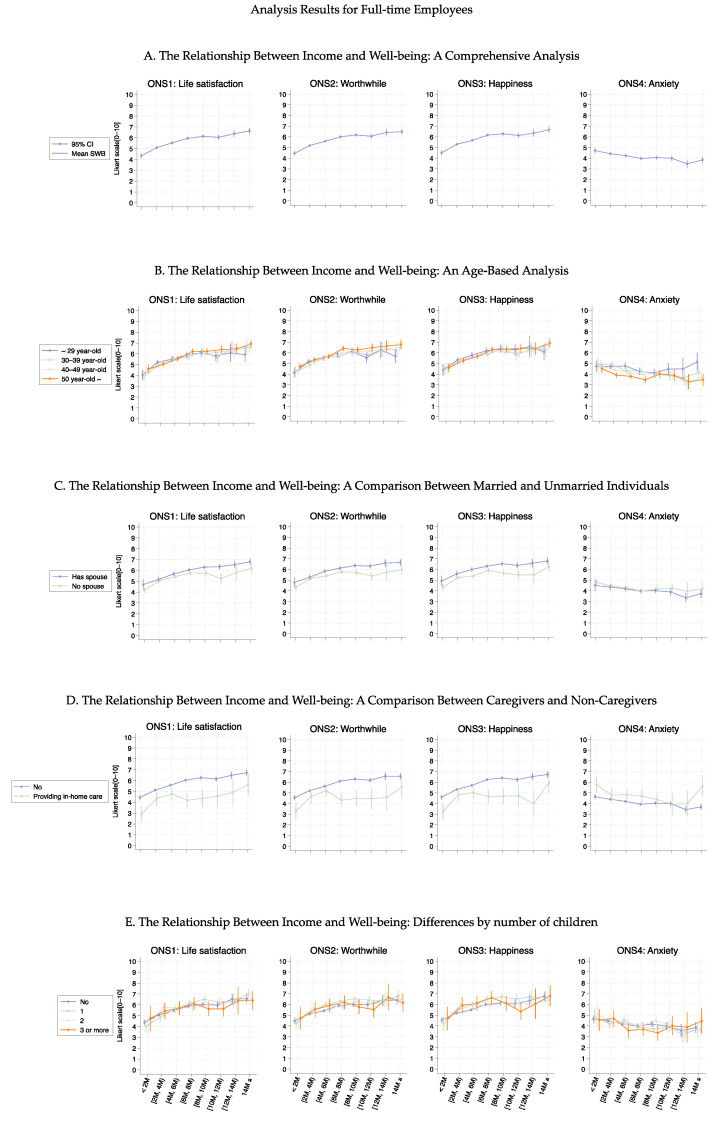
Curves between household income and well-being by basic characteristics among full-time employees.

**Figure 6 healthcare-13-00240-f006:**
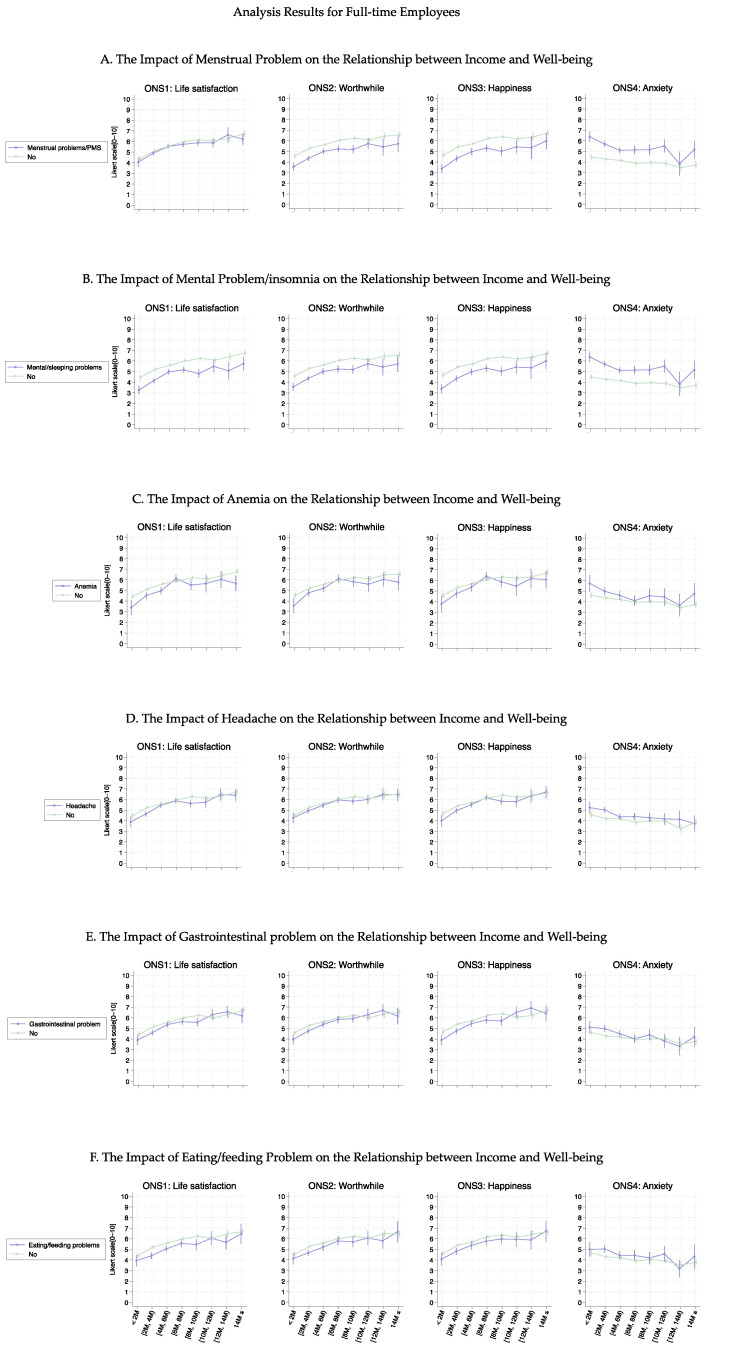
Curves between household income and well-being by female-specific health conditions among full-time employees.

**Figure 7 healthcare-13-00240-f007:**
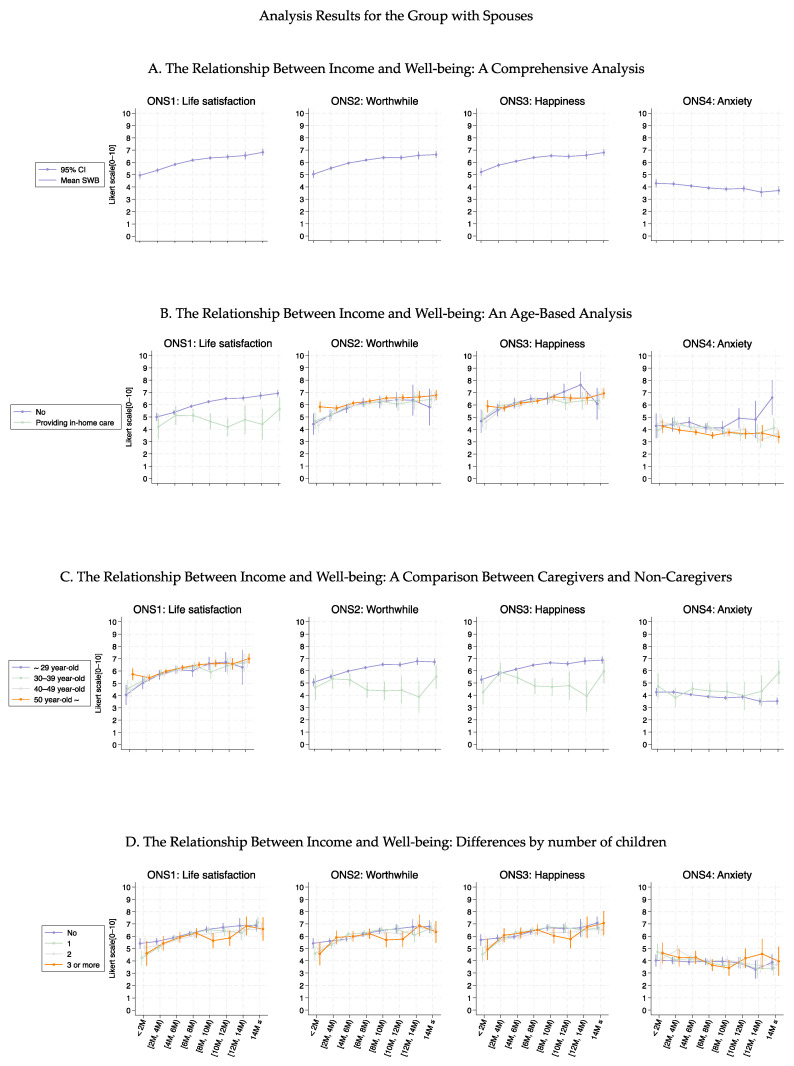
Curves between household income and well-being by basic characteristics among those with a spouse.

**Figure 8 healthcare-13-00240-f008:**
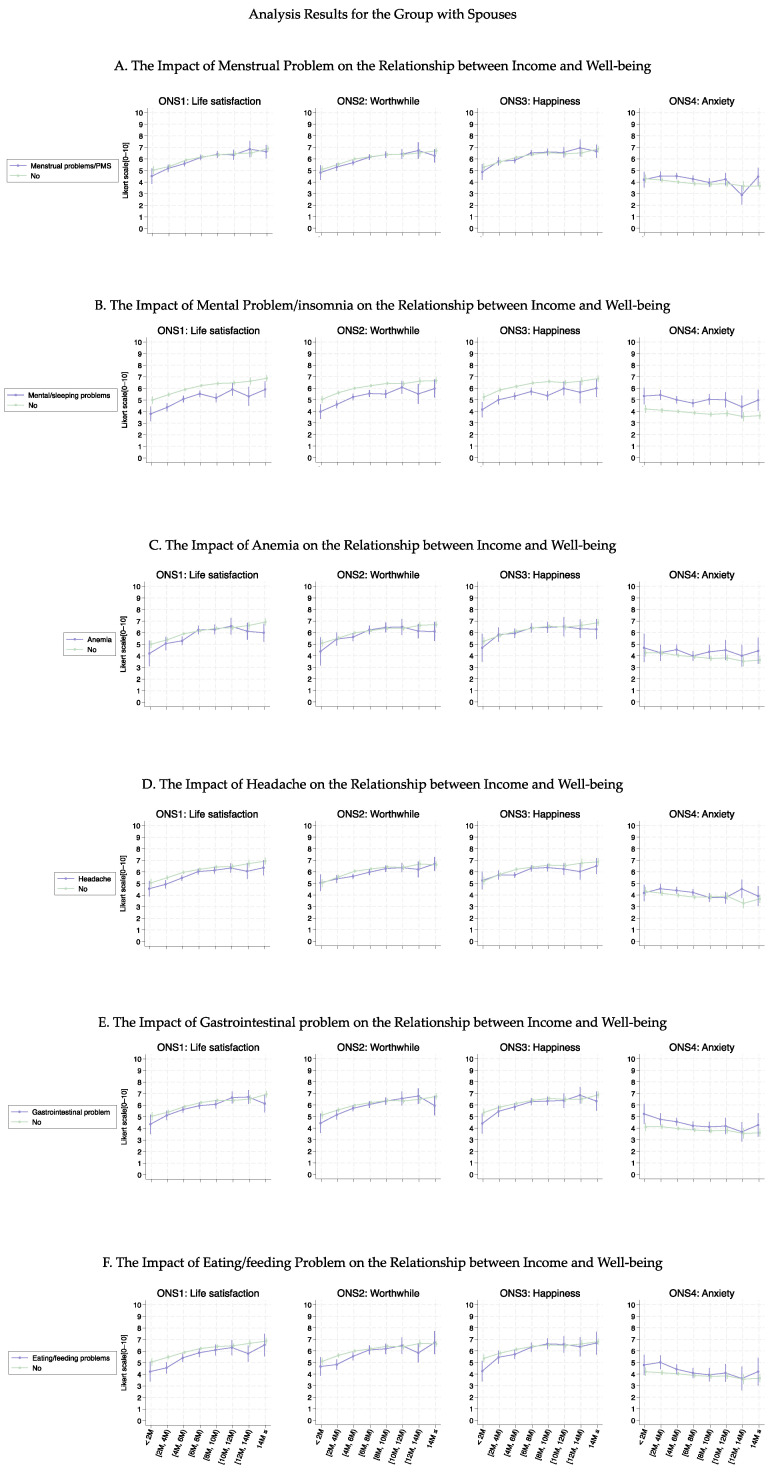
Curves between household income and well-being by female-specific health conditions among those with a spouse.

**Figure 9 healthcare-13-00240-f009:**
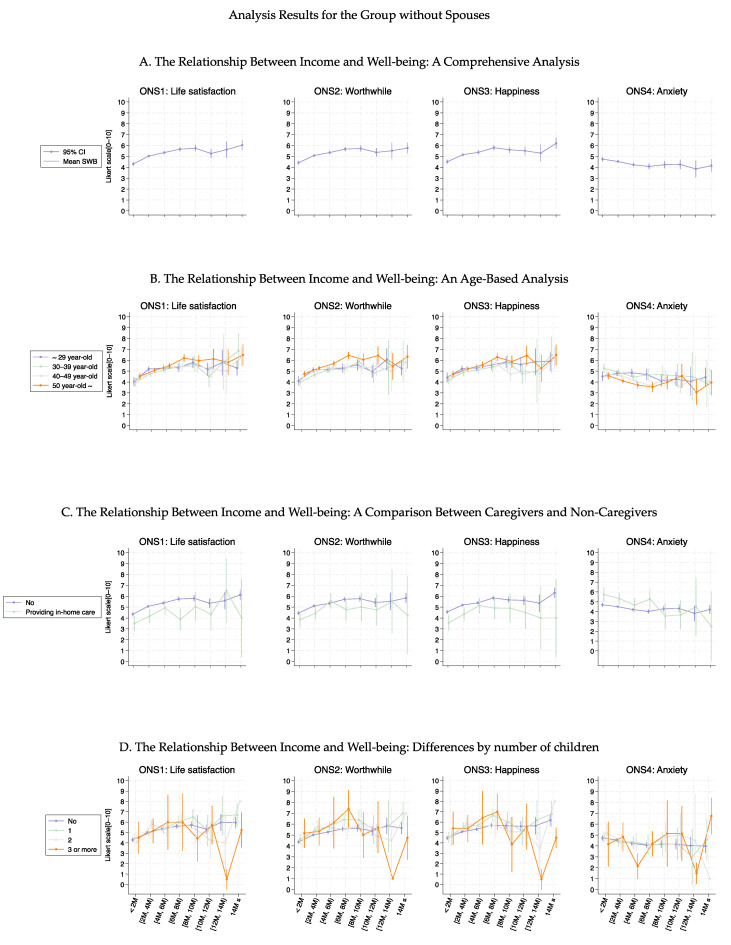
Curves between household income and well-being by basic characteristics among those without a spouse.

**Figure 10 healthcare-13-00240-f010:**
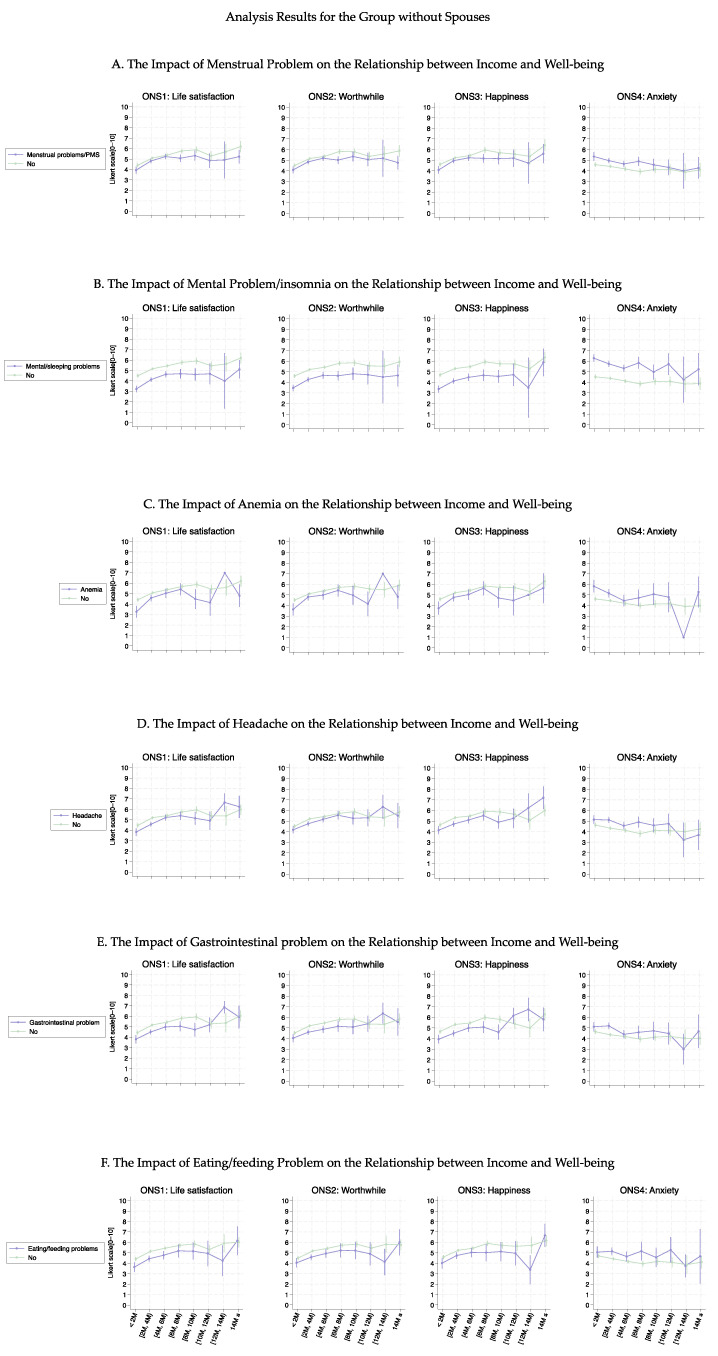
Curves between household income and well-being by female-specific health conditions among those without a spouse.

**Table 1 healthcare-13-00240-t001:** Basic participant characteristics.

	**Household Annual Income**									
	<2 M	2–4 M	4–6 M	6–8 M	8–10 M	10–12 M	12–14 M	14 M>	Total	*p*-value
N (%)	1103 (11.03)	2430 (24.30)	2356 (23.56)	1741 (17.41)	1159 (11.59)	586 (5.86)	247 (2.47)	378 (3.78)	10,000	
Age group (%)										<0.001
20s	163 (14.78)	518 (21.32)	397 (16.85)	297 (17.06)	184 (15.88)	88 (15.02)	25 (10.12)	42 (11.11)	1714 (17.14)	
30s	223 (20.22)	443 (18.23)	509 (21.60)	370 (21.25)	243 (20.97)	116 (19.80)	55 (22.27)	55 (14.55)	2014 (20.14)	
40s	257 (23.30)	582 (23.95)	640 (27.16)	483 (27.74)	331 (28.56)	180 (30.72)	79 (31.98)	124 (32.80)	2676 (26.76)	
50s	355 (32.18)	694 (28.56)	621 (26.36)	485 (27.86)	343 (29.59)	177 (30.20)	73 (29.55)	129 (34.13)	2877 (28.77)	
60s and above	105 (9.52)	193 (7.94)	189 (8.02)	106 (6.09)	58 (5.00)	25 (4.27)	15 (6.07)	28 (7.41)	719 (7.19)	
Occupation (%)										<0.001
Full-time job	487 (44.15)	1576 (64.86)	1379 (58.53)	1134 (65.13)	829 (71.53)	461 (78.67)	194 (78.54)	315 (83.33)	6375 (63.75)	
Part-time job	616 (55.85)	854 (35.14)	977 (41.47)	607 (34.87)	330 (28.47)	125 (21.33)	53 (21.46)	63 (16.67)	3625 (36.25)	
Final academic background (%)										<0.001
Junior high graduate	46 (4.17)	45 (1.85)	35 (1.49)	13 (0.75)	6 (0.52)	3 (0.51)	1 (0.40)	1 (0.26)	150 (1.50)	
Senior high graduate	431 (39.08)	851 (35.02)	620 (26.32)	431 (24.76)	213 (18.38)	91 (15.53)	32 (12.96)	39 (10.32)	2708 (27.08)	
College graduate	159 (14.42)	371 (15.27)	411 (17.44)	277 (15.91)	179 (15.44)	90 (15.36)	35 (14.17)	49 (12.96)	1571 (15.71)	
Vocational school	194 (17.59)	463 (19.05)	420 (17.83)	312 (17.92)	197 (17.00)	70 (11.95)	28 (11.34)	22 (5.82)	1706 (17.06)	
University undergraduate	255 (23.12)	670 (27.57)	818 (34.72)	666 (38.25)	527 (45.47)	299 (51.02)	127 (51.42)	212 (56.08)	3574 (35.74)	
Graduate school	18 (1.63)	30 (1.23)	52 (2.21)	42 (2.41)	37 (3.19)	33 (5.63)	24 (9.72)	55 (14.55)	291 (2.91)	
Spouse (%)										<0.001
Yes	273 (24.75)	792 (32.59)	1394 (59.17)	1269 (72.89)	879 (75.84)	448 (76.45)	201 (81.38)	290 (76.72)	5546 (55.46)	
No	830 (75.25)	1638 (67.41)	962 (40.83)	472 (27.11)	280 (24.16)	138 (23.55)	46 (18.62)	88 (23.28)	4454 (44.54)	
Number of children (%)										<0.001
0	795 (72.08)	1741 (71.65)	1309 (55.56)	816 (46.87)	484 (41.76)	241 (41.13)	86 (34.82)	166 (43.92)	5638 (56.38)	
1	179 (16.23)	389 (16.01)	516 (21.90)	452 (25.96)	323 (27.87)	156 (26.62)	65 (26.32)	86 (22.75)	2166 (21.66)	
2	89 (8.07)	221 (9.09)	405 (17.19)	364 (20.91)	269 (23.21)	139 (23.72)	69 (27.94)	93 (24.60)	1649 (16.49)	
3	35 (3.17)	62 (2.55)	107 (4.54)	99 (5.69)	64 (5.52)	44 (7.51)	21 (8.50)	25 (6.61)	457 (4.57)	
4	4 (0.36)	13 (0.53)	17 (0.72)	10 (0.57)	12 (1.04)	5 (0.85)	4 (1.62)	2 (0.53)	67 (0.67)	
5≤	1 (0.09)	4 (0.16)	2 (0.08)	0 (0.00)	7 (0.60)	1 (0.17)	2 (0.81)	6 (1.59)	23 (0.23)	
Number of cohabitants who require care (%)										0.070
0	1043 (94.56)	2324 (95.64)	2241 (95.12)	1664 (95.58)	1087 (93.79)	550 (93.86)	230 (93.12)	351 (92.86)	9490 (94.90)	
1	51 (4.62)	99 (4.07)	108 (4.58)	70 (4.02)	64 (5.52)	31 (5.29)	14 (5.67)	23 (6.08)	460 (4.60)	
2≤	9 (0.82)	7 (0.29)	7 (0.30)	7 (0.40)	8 (0.69)	5 (0.85)	3 (1.21)	4 (1.06)	50 (0.50)	
Menstrual problems (%)										<0.001
No	827 (74.98)	1690 (69.55)	1648 (69.95)	1233 (70.82)	851 (73.43)	443 (75.60)	197 (79.76)	289 (76.46)	7178 (71.78)	
Yes	276 (25.02)	740 (30.45)	708 (30.05)	508 (29.18)	308 (26.57)	143 (24.40)	50 (20.24)	89 (23.54)	2822 (28.22)	
Mental health problems/insomnia (%)										<0.001
No	805 (72.98)	1832 (75.39)	1845 (78.31)	1411 (81.05)	959 (82.74)	482 (82.25)	208 (84.21)	316 (83.60)	7858 (78.58)	
Yes	298 (27.02)	598 (24.61)	511 (21.69)	330 (18.95)	200 (17.26)	104 (17.75)	39 (15.79)	62 (16.40)	2142 (21.42)	
Anemia (%)										0.633
No	996 (90.30)	2162 (88.97)	2086 (88.54)	1532 (88.00)	1035 (89.30)	528 (90.10)	222 (89.88)	336 (88.89)	8897 (88.97)	
Yes	107 (9.70)	268 (11.03)	270 (11.46)	209 (12.00)	124 (10.70)	58 (9.90)	25 (10.12)	42 (11.11)	1103 (11.03)	
Headaches (%)										0.006
No	834 (75.61)	1800 (74.07)	1735 (73.64)	1313 (75.42)	900 (77.65)	459 (78.33)	190 (76.92)	308 (81.48)	7539 (75.39)	
Yes	269 (24.39)	630 (25.93)	621 (26.36)	428 (24.58)	259 (22.35)	127 (21.67)	57 (23.08)	70 (18.52)	2461 (24.61)	
Gastrointestinal problems (%)										0.007
No	885 (80.24)	1955 (80.45)	1898 (80.56)	1419 (81.5)	975 (84.12)	503 (85.84)	198 (80.16)	319 (84.39)	8152 (81.52)	
Yes	218 (19.76)	475 (19.55)	458 (19.44)	322 (18.5)	184 (15.88)	83 (14.16)	49 (19.84)	59 (15.61)	1848 (18.48)	
Eating/feeding problems (%)										0.036
No	946 (85.77)	2061 (84.81)	2038 (86.5)	1523 (87.48)	1019 (87.92)	516 (88.05)	215 (87.04)	340 (89.95)	8658 (86.58)	
Yes	157 (14.23)	369 (15.19)	318 (13.50)	218 (12.52)	140 (12.08)	70 (11.95)	32 (12.96)	38 (10.05)	1342 (13.42)	

**Table 2 healthcare-13-00240-t002:** Estimated parameters of Tobit regression model for ONS1-4.

		**ONS1: Life Satisfaction**		**ONS2: Worthwile**		**ONS3: Happiness**		**ONS4: Anxiety**	
		Coefficients (95% CIs)	*p*-value	Coefficients (95% CIs)	*p*-value	Coefficients (95% CIs)	*p*-value	Coefficients (95% CIs)	*p*-value
Intercept		4.97 (4.57, 5.37)	0.000	4.53 (4.14, 4.92)	0.000	4.70 (4.26, 5.13)	0.000	4.10 (3.59, 4.61)	0.000
Income		0.22 (0.13, 0.31)	0.000	0.23 (0.15, 0.32)	0.000	0.27 (0.17, 0.37)	0.000	−0.02 (−0.14, 0.09)	0.686
Age group									
	20s	Ref. -	-	Ref. -	-	Ref. -	-	Ref. -	-
	30s	−0.35 (−0.71, 0.00)	0.053	−0.41 (−0.76, −0.06)	0.021	−0.18 (−0.57, 0.21)	0.362	0.39 (−0.07, 0.85)	0.094
	40s	−0.03 (−0.37, 0.31)	0.865	0.00 (−0.34, 0.33)	0.984	0.01 (−0.36, 0.38)	0.963	0.02 (−0.41, 0.46)	0.916
	50s	0.00 (−0.34, 0.34)	1.000	0.34 (0.00, 0.67)	0.047	0.10 (−0.27, 0.47)	0.595	−0.20 (−0.64, 0.23)	0.359
	≥60s	0.39 (−0.07, 0.86)	0.099	0.89 (0.43, 1.35)	0.000	0.59 (0.07, 1.10)	0.025	−0.61 (−1.21, −0.01)	0.047
Marital status									
	No	Ref. -	-	Ref. -	-	Ref. -	-	Ref. -	-
	Yes	0.48 (0.24, 0.72)	0.000	0.49 (0.25, 0.73)	0.000	0.67 (0.40, 0.94)	0.000	−0.40 (−0.71, −0.08)	0.013
Providing in-home care									
	No	Ref. -	-	Ref. -	-	Ref. -	-	Ref. -	-
	Yes	−0.37 (−0.83, 0.10)	0.125	−0.06 (−0.52, 0.39)	0.785	−0.29 (−0.80, 0.22)	0.269	0.39 (−0.21, 0.99)	0.206
Number of children		0.03 (−0.09, 0.16)	0.599	0.21 (0.09, 0.34)	0.001	0.15 (0.01, 0.29)	0.038	0.06 (−0.10, 0.23)	0.459
Menstrual problems									
	No	Ref. -	-	Ref. -	-	Ref. -	-	Ref. -	-
	Yes	0.22 (−0.06, 0.49)	0.120	0.28 (0.01, 0.54)	0.041	0.18 (−0.12, 0.47)	0.245	0.04 (−0.31, 0.38)	0.837
Mental health problems/ insomnia									
	No	Ref. -	-	Ref. -	-	Ref. -	-	Ref. -	-
	Yes	−1.27 (−1.54, −0.99)	0.000	−1.29 (−1.56, −1.03)	0.000	−1.51 (−1.81, −1.21)	0.000	1.90 (1.55, 2.26)	0.000
Anemia									
	No	Ref. -	-	Ref. -	-	Ref. -	-	Ref. -	-
	Yes	−0.11 (−0.47, 0.25)	0.539	0.08 (−0.27, 0.44)	0.638	0.16 (−0.24, 0.55)	0.443	−0.14 (−0.61, 0.32)	0.542
Headaches									
	No	Ref. -	-	Ref. -	-	Ref. -	-	Ref. -	-
	Yes	−0.21 (−0.47, 0.06)	0.123	0.02 (−0.24, 0.28)	0.861	0.02 (−0.27, 0.32)	0.867	0.07 (−0.27, 0.41)	0.689
Gastrointestinal problems									
	No	Ref. -	-	Ref. -	-	Ref. -	-	Ref. -	-
	Yes	−0.16 (−0.44, 0.13)	0.283	−0.29 (−0.57, −0.01)	0.044	−0.43 (−0.74, −0.12)	0.007	0.24 (−0.12, 0.61)	0.194
Eating/feeding problems									
	No	Ref. -	-	Ref. -	-	Ref. -	-	Ref. -	-
	Yes	−0.46 (−0.78, −0.15)	0.004	−0.29 (−0.60, 0.02)	0.071	−0.24 (−0.59, 0.11)	0.184	0.10 (−0.31, 0.51)	0.624
Income × Age group									
	Income × 20s	Ref. -	-	Ref. -	-	Ref. -	-	Ref. -	-
	Income × 30s	0.09 (−0.01, 0.19)	0.072	0.09 (0.00, 0.19)	0.058	−0.01 (−0.12, 0.09)	0.795	−0.16 (−0.29, −0.04)	0.012
	Income × 40s	0.03 (−0.06, 0.12)	0.511	0.03 (−0.06, 0.12)	0.560	−0.05 (−0.15, 0.06)	0.380	−0.14 (−0.26, −0.02)	0.021
	Income × 50s	0.06 (−0.04, 0.15)	0.225	0.01 (−0.08, 0.10)	0.852	−0.03 (−0.13, 0.07)	0.566	−0.12 (−0.24, 0.00)	0.050
	Income × ≥60s	0.04 (−0.09, 0.17)	0.527	−0.04 (−0.16, 0.09)	0.587	−0.09 (−0.24, 0.05)	0.192	−0.08 (−0.24, 0.09)	0.369
Income × Marital status									
	Income × No	Ref. -	-	Ref. -	-	Ref. -	-	Ref. -	-
	Income × Yes	0.02 (−0.05, 0.09)	0.537	0.02 (−0.05, 0.08)	0.563	0.02 (−0.05, 0.10)	0.503	0.05 (−0.04, 0.13)	0.266
Income × Providing in-home care									
	Income × No	Ref. -	-	Ref. -	-	Ref. -	-	Ref. -	-
	Income × Yes	−0.19 (−0.31, −0.07)	0.001	−0.25 (−0.36, −0.14)	0.000	−0.19 (−0.32, −0.07)	0.003	0.02 (−0.12, 0.17)	0.746
Income × Number of children		−0.02 (−0.05, 0.01)	0.142	−0.04 (−0.07, −0.01)	0.005	−0.04 (−0.07, −0.01)	0.024	0.00 (−0.03, 0.04)	0.816
Income × Menstrual problems									
	Income × No	Ref. -	-	Ref. -	-	Ref. -	-	Ref. -	-
	Income × Yes	−0.02 (−0.09, 0.05)	0.581	−0.05 (−0.12, 0.02)	0.186	−0.01 (−0.09, 0.07)	0.826	0.00 (−0.10, 0.09)	0.917
Income × Mental health problems/insomnia									
	Income × No	Ref. -	-	Ref. -	-	Ref. -	-	Ref. -	-
	Income × Yes	0.06 (−0.02, 0.13)	0.148	0.08 (0.01, 0.16)	0.027	0.08 (0.00, 0.17)	0.043	−0.09 (−0.19, 0.00)	0.059
Income × Anemia									
	Income × No	Ref. -	-	Ref. -	-	Ref. -	-	Ref. -	-
	Income × Yes	0.01 (−0.08, 0.11)	0.802	−0.01 (−0.11, 0.08)	0.777	−0.02 (−0.13, 0.08)	0.651	0.04 (−0.08, 0.16)	0.504
Income × Headaches									
	Income × No	Ref. -	-	Ref. -	-	Ref. -	-	Ref. -	-
	Income × Yes	0.03 (−0.04, 0.10)	0.434	0.00 (−0.07, 0.07)	0.961	−0.02 (−0.10, 0.06)	0.569	−0.02 (−0.11, 0.07)	0.701
Income × Gastrointestinal problems									
	Income × No	Ref. -	-	Ref. -	-	Ref. -	-	Ref. -	-
	Income × Yes	0.02 (−0.06, 0.10)	0.614	0.05 (−0.02, 0.13)	0.166	0.08 (0.00, 0.16)	0.062	−0.03 (−0.13, 0.07)	0.569
Income × Eating/Feeding problems									
	Income × No	Ref. -	-	Ref. -	-	Ref. -	-	Ref. -	-
	Income × Yes	0.06 (−0.03, 0.15)	0.164	0.05 (−0.03, 0.14)	0.229	0.07 (−0.03, 0.16)	0.156	−0.01 (−0.12, 0.10)	0.894

## Data Availability

Dataset available upon request from the authors.

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
