# Peer review of "Association Between Income and Well-Being Among Working Women in Japan"

_healthcare, 2025, doi:10.3390/healthcare13030240_

Round 1
Reviewer 1 Report
Comments and Suggestions for Authors
I summarise below some aspects that require revisions:
Sample Representation Issues: The sampling methodology is described as "quota-based random," yet significant limitations emerge from the unequal distribution of participants across income brackets and age groups. This directly compromises the validity of the conclusions, particularly regarding higher-income groups and older participants. The discussion merely acknowledges these limitations without offering substantial mitigation strategies. The authors must address how this impacts the generalisability of the findings and consider techniques such as post-stratification weighting to correct these imbalances.
Superficial Policy Recommendations: The study identifies key issues affecting women’s well-being. However, the policy recommendations remain disappointingly vague. For example, the call for "economic and health-focused policies" is generic and lacks specificity. How should policymakers prioritise interventions like childcare subsidies, workplace flexibility, or nutritional programmes? The authors fail to leverage their findings into actionable strategies, leaving the practical value of the study underdeveloped.
Weak Analysis of Interaction Effects: The regression results highlight significant interaction effects, such as "Income×Providing in-home care," but the discussion offers no in-depth analysis of why these relationships exist or their broader implications. For instance, the findings on caregiving responsibilities (p < 0.001) could have explored how cultural expectations and structural barriers uniquely affect Japanese women. This omission undermines the study's ability to contextualise its data meaningfully.
Underwhelming Visual Presentation: Figures 1 and 2 fail to communicate the most critical findings effectively. The lack of annotations and poor stratification of key variables, such as employment type and marital status, make the visual data challenging to interpret. Additionally, relegating essential plots, like those comparing part-time and full-time workers, to supplementary files diminishes their impact. Improving figure clarity and integrating them into the main text is essential.
Overlooked Ethical Considerations: The authors mention ethical approval but do not sufficiently address the ethical implications of offering e-commerce points as an incentive for participation. This may have biased the sample by disproportionately attracting lower-income respondents, skewing the data. A critical reflection on this potential bias is absent, weakening the study's credibility.
Reviewer 2 Report
Comments and Suggestions for Authors
Thank you for the opportunity to read the paper and share my comments with the Authors. In my opinion, the paper requires corrections.
The title of the paper should be corrected. It is not necessary to provide "among 10,000 working women" in the title.
The literature review should be expanded. The authors referred to only 26 literature items in total.
Tables are too large. The most important elements should be presented in the main text of the paper, and the others should be given in appendices. Figures are too small, difficult to read.
The authors refer to the results to a small extent, they do not comment on the results.
The Conclusions section should be expanded. In the current version, it is only three sentences!!!
Reviewer 3 Report
Comments and Suggestions for Authors
Please refer to the attached file for my comments. I urge the authors to incorporate all these points in detail in a revised version.
----
Referee Report: Nonlinear association between income and well-being among 10,000 working women
Overall assessment. This paper addresses a very important question. The methodology employed by the authors (survey evidence) is very appropriate and the overall exposition of the paper is generally ok. Despite the promising outlook, the paper is not publishable in its present form. The main reservation I have with the manuscript is that it reads as if the findings of the paper were a mere tautology.
Nevertheless, this is something that can be definitely addressed in a diligent revision through a combination of expositional edits and additional empirical tests. In the following paragraphs, I discuss what is the main limitation of the present version and how the authors can address my concerns and write a publishable piece. Hence, I urge the authors to incorporate all my comments in detail, in order to improve the scholarly quality of their work.
Interpretation of the findings. The main findings of the paper can be construed as follows: (1) well-being increases with income and (2) the positive slope generally decreases as we increase the level of income (i.e., well-being as a function of income is a concave function).
Well, at the first glance, the reader can interpret these findings as being just the natural consequence of the canonical utility theory, wherein utility is increasing in most consumption goods that make individuals “better off” (which is certainty the case of income). Moreover, the non-linearity (concavity) in question can be interpreted simply as corroborating evidence that marginal utility decreases when consumption increases. Differently stated, the effect of one extra dollar (or yen) on the utility (or well-being) of an individual whose income is 100 dollars (or yens) is larger than the effect of the same extra dollar (or yen) on the utility of an individual whose income is 100,000 dollars (or yens).
In other words, an overly critical reader can at first undermine the contribution of the manuscript by assuming that it is just providing evidence of a very well-established (canonical) fact in choice theory: that well-behaved preferences (i.e., preferences that can be represented by a utility function) display decreasing marginal utility.
While I do not share this view (i.e., I really believe that the substance of this paper is worth documenting), it is important to underscore that the results corroborate canonical choice-theory models with utility functions concave in wealth (and the concavity itself indicates the degree of risk aversion of individuals), the main contribution of the paper is shedding light on the specific magnitudes of the effect of income on well-being, as well as other parametric and non-parametric features of the relationship between income and well-being.
Additional analysis. Having discussed the aforementioned points, the authors should show that the positive association between well-being and income is not subject to estimation biases due to omitted correlated factors. To do so, the authors should perform an analysis akin to the one of Dantas et al. (2023) (refer to Table 11 and Table 12 of Dantas et al., 2023). To estimate parametric bounds for the coefficient β of income, the authors should do as follows:
â‹„ Estimate an OLS regression of well-being on income without the inclusion of any control variable (nor fixed effects). Critically, the authors should collect the coefficient estimate of income (βË™) and the Rsquared of this simplified regression (RË™2).
â‹„ Estimate an OLS regression of the “augmented” model, which should include some of the individual control variables considered in the paper. This regression should give another coefficient estimate of income (β˜) and another value of R-squared (R˜2).
â‹„ Estimate the parametric bound following the equation in Dantas et al. (2023): β∗ = β˜ - (βË™-β˜)( R˜R2-max 2RË™ 2-R˜2).
If β∗ is also positive, then the authors have a compelling point for the coefficient estimate (elasticity) to be positive even when selection biases are taken into account.
Contextualization. I think the paper would also benefit from some contextualization regarding the generalizability of their findings. For example, the concavity of utility/well-being as a function of income is a proxy for individuals’ risk aversion (the more concave, the more risk averse the individual is). Hence, would results be different if applied to other groups of individuals (i.e., other groups based on gender or nationality)? Similarly, would results be different if the survey is conducted at a different time? After all, in the aftermath of a pandemic, it is plausible that individuals are more risk-averse than their historical average?
References
Dantas, M., K. J. Merkley, and F. B. G. Silva (2023). Government Guarantees and Banks’ Income Smoothing. Journal of Financial Services Research 63(2), 123–173. DOI: https://doi.org/10.1007/s10693-023-00398-3.

Round 2
Reviewer 1 Report
Comments and Suggestions for Authors
The authors addressed all previous concerns. I am glad to see that the article is now up to the standards.
Author Response
I am very grateful for your time and effort in providing a lot of insightful comments.
Thank you very much.
Reviewer 2 Report
Comments and Suggestions for Authors
The authors have responded in detail to the comments I made in the paper review. They have made corrections to the paper. In my opinion, the paper in the presented version after corrections is much better and clearly presents the problem being studied.
The paper is suitable for printing in the Journal, in my opinion.
Author Response

(The authors gave the same response as above.)

Reviewer 3 Report
Comments and Suggestions for Authors
I thank the authors for responding to my comments with professionalism and for their diligent attempts to address all the points raised in the previous round.
While I concur with the authors' arguments that the Oster (2019) test relies on the OLS assumptions, I believe it can still bolster the internal validity of their study if the authors perform a robustness test akin to the one in Dantas et al (2023). Again, I ask the authors to refer to Table 11 and Table 12 of Dantas et al (2023), where the parametric bounds are estimated based on the coefficients of two regressions: one with a simplified (smaller) set of controls and another with an augmented set of controls. Both regressions require OLS estimations but, regardless of the methodology, their value stems from the parametric bounds having the same sign as the baseline estimates.
Applying the approach of Dantas et al (2023) in the context of the present work, I am not suggesting that the authors change the entire analysis from Tobit to OLS estimates. In fact, Tobit estimates are very appropriate. My point is that the robustness test itself can be estimated with the two OLS regressions akin to Dantas et al (2023) while the core analysis still remains following the Tobit estimates. The authors can provide the justification for the use of OLS estimations to conduct the robustness test proposed by Dantas et al (2023) based on the arguments they provided in their response memo to the comments raised in the previous round.
Dantas, M.M., Merkley, K.J. and Silva, F.B., 2023. Government guarantees and banks’ income smoothing. Journal of Financial Services Research, 63(2), pp.123-173.
Oster, E., 2019. Unobservable selection and coefficient stability: Theory and evidence. Journal of Business & Economic Statistics, 37(2), pp.187-204.
